

# Geomorphic risk maps for river migration using probabilistic modeling - a framework

Brayden Noh[1*], Omar Wani[1,2*], Kieran B.J. Dunne[1,3], and Michael P. Lamb[1]

[1]Division of Geological and Planetary Sciences, California Institute of Technology, Pasadena, CA 91125, USA
[2]Department of Civil and Urban Engineering, New York University, Brooklyn, NY 11201, USA
[3]Faculty of Civil Engineering and Geosciences, Delft University of Technology, Delft, 2628 CN, The Netherlands
[*]These authors contributed equally to this work.

**Correspondence:** Omar Wani (omarwani@nyu.edu)

**Abstract.** Lateral migration of meandering rivers poses erosional risks to human settlements, roads, and infrastructure in alluvial floodplains. While there is a large body of scientific literature on the dominant mechanisms driving river migration, it is still not possible to accurately predict river meander evolution over multiple years. This is in part because we don't fully understand the relative contribution of each mechanism and because deterministic mathematical models are not equipped to
account for stochasticity in the system. Besides, uncertainty due to model-structure deficits and unknown parameter values remains. For a more reliable assessment of risks, we, therefore, need probabilistic forecasts. Here, we present a workflow to generate geomorphic risk maps for river migration using probabilistic modeling. We start with a simple geometric model for river migration, where nominal migration rates increase with local and upstream curvature. We then account for model structure deficits using smooth random functions. Probabilistic forecasts for river channel position over time are generated by monte carlo
runs using a distribution of model parameter values inferred from satellite data. We provide a recipe for parameter inference within the Bayesian framework. We demonstrate that such risk maps are relatively more informative in avoiding false negatives, which can be both detrimental and costly, in the context of assessing erosional hazards due to river migration. Our results show that with longer prediction time horizons, the spatial uncertainty of erosional hazard within the entire channel belt increases - with more geographical area falling within 25% < probability < 75%. However, forecasts also become more confident about
erosion for regions immediately in the vicinity of the river, especially on its cut-bank side. Probabilistic modeling thus allows us to quantify our degree of confidence - which is spatially and temporally variable - in river migration forecasts. We also note that to increase the reliability of these risk maps, we need to describe the first-order dynamics in our model to a reasonable degree of accuracy, and simple geometric models do not always possess such accuracy.

## 1 Introduction

Meandering rivers migrate in their alluvial plain due to differential erosion and deposition along the outer and inner banks. Among other processes, this migration primarily happens because the shear stresses exerted on the cut bank are relatively high and the shear stresses on the point bar are relatively low, leading to erosion on one side and deposition on another (Dietrich and Smith, 1983; Parker et al., 2010; Phillips et al., 2022). This is, in turn, a consequence of spatial divergence and convergence of





the sediment flux, respectively, caused by spatial acceleration and deceleration of the flow. There is evidence that larger rivers,
when averaged globally, migrate faster than small rivers, and in some cases, such migration rates reach tens to hundreds of
meters per year (Langhorst and Pavelsky, 2023). As a large fraction of the human population lives in close proximity to rivers,
such rates of bank erosion pose risks to infrastructure and communities in the vicinity of these rivers (Wu et al., 2023; Jarriel
et al., 2021).

Both local and nonlocal mathematical theories have been postulated to understand the dynamics of river migration (Board
et al., 2004; Güneralp et al., 2012; Bogoni et al., 2017). The spatially localized theories focus on the erodibility of the bank
material versus the hydraulic and gravitational stresses that induce erosion. The dominant processes that cause such an aggre-
gate bank retreat are: nearbank and overbank flow, seepage flows, fluctuations in soil pore pressure, and intermittent collapse
of overhanging slump blocks (Zhao et al., 2022). Previous studies have explored the effects of floodplain heterogeneity and
substrate strength on meandering river planform and migration (Güneralp and Rhoads, 2011; Limaye and Lamb, 2014; Bogoni
et al., 2017). In contrast, nonlocal theories are primarily geometric, where the curvature over a channel length is used as a
predictive variable that tells us about the effective erosion rate experienced at various locations (Howard and Knutson, 1984;
Sylvester et al., 2019). These geometric models use a weighted sum of upstream curvatures to forecast the effective migration
at a location along the channel. Geomorphic and hydrologic features of the alluvium and river are represented by fitting the
model parameters to the observed migration. The benefit of these geometric models is their computational economy. Local
models rely on equations of motion to calculate the velocity and depth fields, which in turn allow for the calculation of shear
stress distribution on the banks. This shear stress field can then be used to estimate erosion and deposition rates. However, such
a spatially distributed physics-based approach comes with large data needs and a heavy computational burden. The geometric
models, on the other hand, rely on the emergent behavior of river migration. These models can be run over and over, allowing
us to compute various statistics of the forecast (Posner and Duan, 2012). This computational efficiency is especially helpful in
making predictions in the probabilistic framework.

Along with working on the predictive accuracy of earth and environmental science models, over the past few decades, there
has been a growing interest in and acknowledgment of the cascading chain of uncertainties, especially when these models are
used to evaluate risks and engineer solutions. Proper treatment of such uncertainties allows for risk-based decision-making.
However, there is an underlying prerequisite to enable such a shift in decision-making paradigms – that is, the evaluation of
reliable forecast probabilities. And this holds true for river migration modeling as well (Jerolmack, 2011). Due to stochastic
variability (Scheidegger, 1991) - in hydraulic stressors, sediment supply, and mechanical properties of banks - and due to
model structure deficits, deterministic representations are generally only able to capture the first-order dynamics. To assess
the geomorphic risk of the meandering river to its surrounding reaches, we, therefore, propose embedding deterministic river
migration models into a probabilistic framework. This work takes inspiration from risk maps that are offered for other earth-
systems-related hazards like floods, droughts, hurricanes, and earthquakes. The goal of this work is to combine concepts from
probability theory and geomorphic modeling and provide a step-by-step guideline to account for - within the model represen-
tation - various model-related uncertainties and stochasticity in the system. We proceed to constrain parameter uncertainties
using observational data within the Bayesian framework.



There has been previous work on introducing probabilistic analysis to geomorphic modeling by introducing Monte Carlo
simulations over parameter values as well as incorporating distributions of rainstorm intensity, discharge, sediment motion, and
sediment supply (Posner and Duan, 2012; Benda et al., 1998; Benda and Dunne, 1997a, b; Dunne et al., 2016). Here, we wish
to extend that literature and provide a more comprehensive recipe to a) try accounting for model structure deficits by deploying
additive smooth random functions, then b) infer distributions of model parameters from observations, and c) finally, generate
geomorphic risk maps using forward Monte Carlo simulations. For this, we also had to devise an algorithm that determines
whether a modeled channel has passed over a geographical location or not.

In the next sections, we will go over methodological details describing how to make geometric models stochastic and
generate risk maps using inferred parameter distributions. We present the results from our simulation experiments on synthetic
data and on a real case study. After the results, we discuss the advantages of this method over purely deterministic prediction
setups. Finally, we discuss some of the unique challenges in generating risk maps for migrating rivers - growing uncertainty
with increasing prediction horizon, identification of adequate geometric models that capture the first-order dynamics, and the
nonlinearity introduced by cutoffs. We end the manuscript with some generalizable conclusions from this analysis.

## 2 Methods and material

Here, we introduce the recipe to extend a deterministic geometric model of river migration into a stochastic model. We first
motivate river migration as a problem concerning an evolving curve. We introduce a simple geometric model that can guide
channel evolution. We then go through the various additive stochastic terms that can be used to account for model structure
deficits. We propose a novel application of smooth random functions for the river migration dynamics, which is different in
characteristics than other space-time processes like discharge time series or flow velocity fields. Once the stochastic model for
river migration is defined, we prepare the inverse problem by selecting the observations and defining the likelihood function.
We finally introduce a novel counting algorithm as a counting problem for a 1D manifold in a 2D space and explain how to
generate risk maps from the initial and final channel geometry - which is much more efficient than tracking the geometry during
the entirety of evolution time.

### 2.1 Geometric geomorphic models and uncertainty

In geomorphic models, like most predictive models in earth and environmental sciences, we plug the input variables in the
model and get the output. However, the nature of the dependent variable is different - unlike a time series or a geospatial
field, it's a dynamic 1D manifold, when channel evolution is viewed in its totality. In the context of fluvial geomorphology,
we generally assume the channels to be smooth planar curves (a one-dimensional manifold in a real Euclidean plane $\mathbb{R}^2$), and
their forward simulation provides the evolution of this curve given an initial channel geometry.

$$\mathcal{C}_t = \left\{ (x,y) \in \mathbb{R}^2 \mid f(\theta,t), \, \mathcal{C}_0 \right\} \tag{1}$$





Here $\mathcal{C}_t$ are the subset of points in $\mathbb{R}^2$ space representing the channel, and $f(\theta, t)$ represents a deterministic function with
90 parameters $\theta$ that guide the evolution of the channel from its initial state $\mathcal{C}_0$ to $\mathcal{C}_t$ at time $t$ (Fig. 1). These parameter values indirectly reflect the geomorphological and hydrological processes active in each channel reach. We can extend this definition to incorporate stochastic elements in channel evolution. In addition to $f$, which captures the first-order dynamics, we can have an additive stochastic term $g$ to capture the deviations from the average behavior.

$$\mathcal{C}_t = \left\{ (x, y) \in \mathbb{R}^2 \mid f(\theta, t) + g(\psi, s), \mathcal{C}_0 \right\} \tag{2}$$

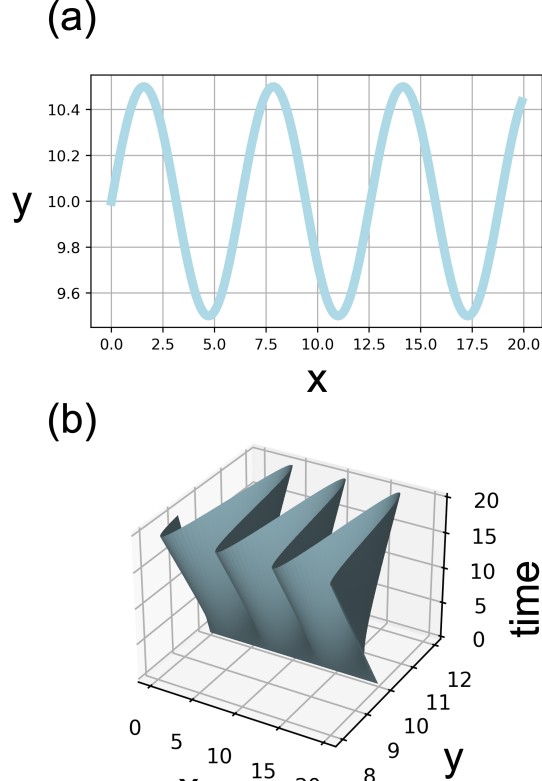

**Figure 1.** This illustrative figure depicts river migration as a problem of evolving planar curves. (a) A river channel can be depicted as a 1D manifold in $\mathbb{R}^2$, assuming it to be the set of points $\mathcal{C}_{t*}$ at an arbitrary time t*, where $x$ and $y$ represent the two spatial dimensions in the alluvium. (b) A river channel can also be depicted as a 2D manifold in $\mathbb{R}^3$, assuming it to be the set of points $\mathcal{C}_{0:t}$, where interval 0 to t is the time of evolution from some arbitrary initial to arbitrary final state.

One of the most popular geometric models used for estimating river migration was proposed by Howard and Knutson (1984). This model works under the assumption that more curved regions of the channel experience more shear stresses and therefore,





on average, migrate more. The migration rate can thus be considered as a compound effect of local curvature and curvature upstream. The effective migration $R_1$ rate is estimated as the weighted sum of local and upstream curvatures:

$$R_1(s) = \Omega R_0(s) + \left[ \Gamma \int\limits_0^\infty R_0(s-\xi)G(\xi)d\xi \right] \left[ \int\limits_0^\infty G(\xi)d\xi \right]^{-1} \tag{3}$$

Where Sylvester et al. (2019) describe the nominal migration rate $R_0$ as the one that is only dependent on the local curvature:

$$R_0 = k_1 W/R \tag{4}$$

$\Omega$ and $\Gamma$ are weighting parameters and (W/R) is the ratio of width to the radius of curvature and $k_1$ is the migration rate constant. They assume the values of $\Omega$ and $\Gamma$ to be –1 and 2.5. In this model, $s$ is the location along the centerline at time t, $\xi$ is the along-channel distance upstream from that location, and $G(\xi)$ is a function for the convolution integral, which weights

the contribution of upstream curvature:

$$G(\xi) = e^{-\alpha\xi} \tag{5}$$

where $\alpha$, a tunable parameter, is a function of a dimensionless friction factor $C_f$ and water depth $D$, and $k$ is a constant that equals 1 :

$$\alpha = 2kC_f/D \tag{6}$$

Sylvester et al. (2019) have developed a python package, MeanderPy, which uses these equations for channel migration and comes with very handy analysis and visualization tools. However, to constrain the channel evolution with site-specific geomorphic properties, we first need to determine the model parameter values, which vary from case to case. Learning about these parameters from observations of the system response is labeled as the inverse problem, where the question being answered is: given an initial and final channel geometry, what values of parameters could have produced these dynamics? This inversion of

parameter values is confounded by noise in the process and model structure deficits.

In deterministic modeling paradigms, for a given input channel geometry and parameter values, a unique output geometry is expected. Observed deviations over and above the deterministic model $f$ are generally neglected. All we are after is the most likely evolution that we expect from a migrating river system. The model calibration, i.e., tuning of its parameters, is carried out using specific optimization metrics like seeking the least squared error by comparing the predicted and observed channel

geometry. To prevent fitting the model to the noise, cross-validation techniques are employed, where observations are split into three categories: calibration, validation, and test. However, within probabilistic paradigms, we are not only interested in the "best guess" about parameter values and model output, but we want to employ the whole distribution of the parameters and, in turn, obtain the entire distribution of channel output; we start with a mathematical framework that can assign probabilities to our modeled system response. Given the inherent nonlinearities in the migrating river systems, we posit using a distribution of

inferred parameter values allows for more reliable forecast, with an explicit consideration for uncertainties.



While geometric geomorphic models lose accuracy due to simplifying assumptions, models that describe the system in greater detail - by incorporating hydraulics and soil mechanics - do not necessarily improve the predictive capability. This is because many other factors play a role. An attempt at adding more subprocesses to the system description also requires that the representation of those subprocesses is correct - i.e., the model is able to represent the true dynamics of the system. Besides, more data is needed to feed these models. So there is a tendency to accumulate errors when we go to more detailed models that resolve various spatial processes. Therefore, geomorphic models tend to have uncertainties associated with them irrespective of the detail with which they try to describe the hydrologic and geomorphic systems.

The specific sources of uncertainty in geomorphic models are as follows: a) Model-structure uncertainty, which arises from the fact that the equations used to describe the evolution of a system like (a meandering channel) are approximate. The model deficits can lead to systematic over or underestimation of the migration rates, or it can also lead to deviations in more random ways. b) The measurement of the input, elevation, and the system response itself suffer from systematic and random deviations. This is called observational uncertainty. These deviations may be stationary in time or show some dependence on external drivers. This uncertainty becomes especially more pronounced when we want to carry out inferences about the system invariants - like its parameters or model structure - using this data. c) The other source of uncertainty is unknown parameter values. Given that many combinations of parameter values, the ones we have not measured, can produce a close fit to the observed migration of the channel geometry to a comparable degree, leading to parameter uncertainty. Parameter uncertainty is usually a result of indeterminacy in the deterministic paradigm, i.e., given an algebraic or differential equation, there are n data points, depending on the nature of the equation, necessary to fully define the system. Having fewer observations then results in parameter uncertainty. In the probabilistic paradigm, parameter uncertainty is a consequence of the fact that different parameter values, once a model is defined, can explain the same observational data with varying probabilities because, as mentioned, observations have errors and models are imperfect. In an ideal world, for example, for fitting a line, two observations with no noise will uniquely define its parameters. However, the presence of random error allows us to fit many lines through the same data.

Given all these cascading chains of uncertainty, there is prudence and value in quantifying them for risk assessments. Risk can be described as a combination of costs associated with various possible outcomes and the associated uncertainty in the realization of those outcomes. Thus, metrics like expected loss try to encapsulate some aspects of risk. However, the underlying requirement to have some faithful quantification of risk is the evaluation of event probabilities. Probabilities can be derived from past frequencies of the phenomena in question or using stochastic models of the process. Various natural hazards are reported in terms of their risk maps. For example, seismic risk maps, weather-related risk maps, and flood risk maps. When forecast variables have a hazard associated with them, most fields rely on the concept of risk. They generate probabilistic forecasts, which in turn are used to make risk assessments.

## 2.2 Extension of the deterministic models into stochastic models

Model structure deficits can be thought of as an aggregation of small deviations that come from several sub-processes not being incorporated in the model equations. We can therefore assume errors due to model structure will follow some parametric





distribution in the limit. The simplest case that is often employed is the normal distribution at each location. However, if we want more structure to the additive term, we will have to upgrade from random variables to random functions, which are called stochastic processes.

**Nonsmooth stochastic processes**

Mathematical models of earth and environmental systems can be considered the sum of a deterministic term and a random variable (Higdon et al., 2004; Wani et al., 2017b, a; Kennedy and O'Hagan, 2001; Wani et al., 2019). If the predicted variable is time or space continuous, we need to borrow more advanced concepts from statistics and use random functions or fields as additive terms. These space/time continuous random functions or fields are referred to as continuous stochastic processes. Reichert and Schuwirth (2012) propose to describe true system response at time $t$ of an environmental system as the sum of a deterministic model and a stochastic process. If we adopt this method, the deterministic model $f(\theta, t)$ is given by the Howard-Knutson model. And the stochastic term $g(\psi)$ can be given, for example, by the Ornstein-Uhlenbeck process $\mathcal{Y}$, defined as:

$$d\mathcal{Y}_x = -\beta \mathcal{Y}_x \cdot dx + \gamma \cdot dW_x \quad : \quad W_{x+\Delta x} - W_x \sim \mathcal{N}(0, \Delta x) \tag{7}$$

Here, $W$ is a Weiner process. Wiener process is the continuous version of the normal noise, where increments within an interval $\Delta x$ are independent, normally distributed, and with variance equal to the interval length. Given its mathematical definition (Eq. 7), the Ornstein–Uhlenbeck process comes out as stationary, Gaussian, and Markovian (Van Kampen, 2007). The solution of the stochastic differential equation above, ie. $\mathcal{Y}_x$, is a continuous random function, which is normally distributed at each spatial location $x$ (Rasmussen and Williams, 2005). As $f(\theta, t)$ maps us to $(x, y)$, we use an additive $g(\psi)$ for $x$ and $y$ each (illustrated in Fig. 2). Various other configurations can be used to randomly perturb the $f(\theta, t)$ model, but we start with the simplest of configurations. Another configuration that is possible: $f(\theta, t) \cdot g(\psi)$. However, the function is not a smooth function, and therefore, if used as $g(\psi)$ does not provide smooth river channels.

**Smooth stochastic processes**

Modeling geomorphic evolution as a stochastic process comes with a unique challenge in the case of migrating rivers - at the scales we are interested, channels appear to be smooth (Fig. 2a). However, smoothness is not a requirement in other space-time processes related to other earth and environmental systems. When we perturb the river channel evolved by a geometric model using gaussian processes (for example, Eq. 7), the final channel geometry is jagged and does not resemble the smooth planar curves we observe at the scales that we are interested in. Fortunately, statisticians have also formulated descriptions of random functions that are smooth and normally distributed at $x$. Filip et al. (2019) elaborate on how to use truncated Fourier series with random coefficients to generate such smooth stochastic processes.

$$\mathcal{Y}_x = a_0 + \sigma\sqrt{2} \sum_{j=1}^{m} \left[ a_j \cos\left(\frac{2\pi j x}{L}\right) + b_j \sin\left(\frac{2\pi j x}{L}\right) \right], \quad m = \lfloor L/\lambda \rfloor \tag{8}$$



where each $a_j$ and each $b_j$ is an independent sample from $\mathcal{N}(0, 1/(2m+1))$ and $\sigma$ is the standard deviation. And $L$ is the domain length of the random function. Depending on the choice of $\lambda$, we can choose the number of sines and cosines we wish to add to generate our final smooth random function. The limit $\lambda \to 0$ generates a gaussian process. If we wish to learn about the parameters, like $m$ of such stochastic processes, we can also try to infer their values from observations.

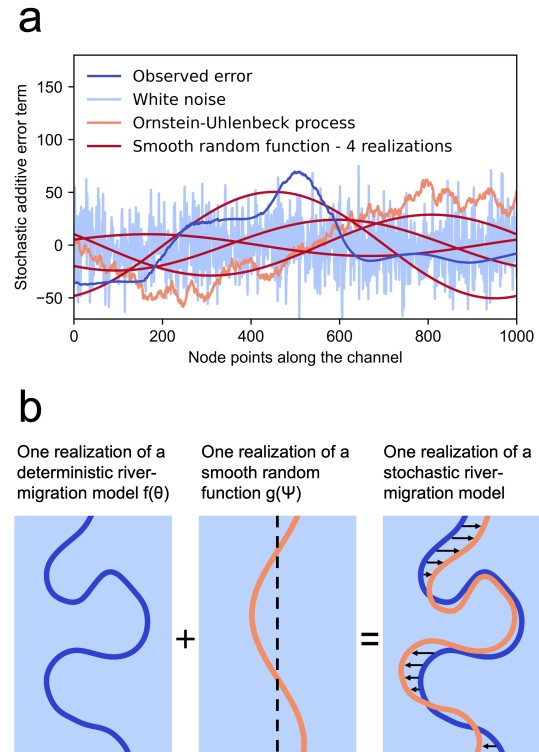

**Figure 2.** This figure demonstrates an additive scheme that allows us to model river migration as a stochastic process. (a) A visualization of additive smooth and nonsmooth random functions and comparing it to one of the realizations of observed error. (b) Producing a stochastic realization of river migration using a deterministic evolution model f($\theta$) and a stochastic additive term g($\psi$). In our formulation, we add two different runs of the smooth noise to the x values and y values each.

## 2.3 Learning from observational data

In the context of channel migration, a likelihood function $p(\mathcal{C}_t \mid \theta, \psi, \mathcal{C}_0)$ is the probability density of observing a certain evolved channel $\mathcal{C}_t$ at time $t$, given the initial geometry $\mathcal{C}_0$, conditional on specific values of $(\theta, \psi)$ that produced it. Additionally, having a prior probability distribution over the model parameters $p(\theta, \psi)$, which can represent physical constraints or expert knowledge, allows us to write the inference of parameters from observations $\mathcal{C}_t$ as a Bayesian problem. The updated parameter distribution, after learning from data, is called the posterior distribution, written as (Korup, 2021):



$$p(\theta, \psi \mid \mathcal{C}_t, \mathcal{C}_0) = \frac{p(\mathcal{C}_t \mid \theta, \psi, \mathcal{C}_0)\, p(\theta, \psi)}{\int p(\mathcal{C}_t \mid \theta, \psi, \mathcal{C}_0)\, p(\theta, \psi)\, d\theta d\psi} \tag{9}$$

This distribution updates our estimate of the parameter values of the model and narrows down the range of the estimate after learning from the observed data (Fig. 3). While using the smooth formulation of additive stochastic processes gives us the advantage of producing more realistic channel geometries, we lose some of the analytical properties that come with gaussian processes. Most importantly, using a gaussian process as an additive stochastic term allows for explicitly evaluating the likelihood function $p(\mathcal{C}_t \mid \theta, \psi, \mathcal{C}_0)$. This isn't straightforward in the case of smooth random functions. However, we can

write an explicit likelihood function on a specific summary statistic of our channel geometry - the coefficients of sines and cosines in the smooth random function, as they are, by definition, normally distributed ($a_j$ and $b_j$ in eq. 8). And we can recover these coefficients from observations using Fourier transform on the residuals i.e. $\mathcal{C}_t^* - f(\theta, t)$. We run the deterministic model $f(\theta)$, and then subtract the deviations in $(x, y)$ of the predicted channel using $f$ from the observed channel. From our construction in equation 2, this difference is distributed as $g(\psi)$. Fourier transform on one observed realization of $g$, called

$g_{obs}$, yields a function in the frequency space.

$$\hat{f}(\phi) = \int\limits_{-\infty}^{\infty} g_{obs}(s)\, e^{-i2\pi\phi s}\, ds \tag{10}$$

$s$ is the distance along the channel from an arbitrary upstream point. In our current formulation, we add $g$ separately to both x-values and y-values, therefore $s$ becomes $x$ and $y$ in each case. We can now use the fast Fourier transform to get the values of $\hat{f}(\phi)$. Depending on the composition of eq. 8, we expect the maximum values of $2/\mathfrak{size}(g_{obs}(s)) \cdot \mathfrak{Re}(\hat{f}(\phi))$ and

$2/\mathfrak{size}(g_{obs}(s)) \cdot \mathfrak{Im}(\hat{f}(\phi))$ to give us $a_j$ and $b_j$ respectively (here $\mathfrak{Im}()$ is the imaginary part and $\mathfrak{Re}()$ is the real part). We can explicitly calculate the likelihood function for them as we assume to be independent and normally distributed with zero mean and variance $\psi$. In this study, we use the affine-invariant ensemble sampler for Markov Chain Monte Carlo (Goodman and Weare, 2010), packaged in python as EMCEE by Foreman-Mackey et al. (2013, 2019) to sample from the posterior distribution. Another way to infer parameter values would be using Approximate Bayesian Computation (ABC), where we don't need to

evaluate the likelihood function but only be able to sample from it, i.e., be able to run the stochastic model $f + g$ as forward simulation. We have not analyzed an ABC scheme in this work.





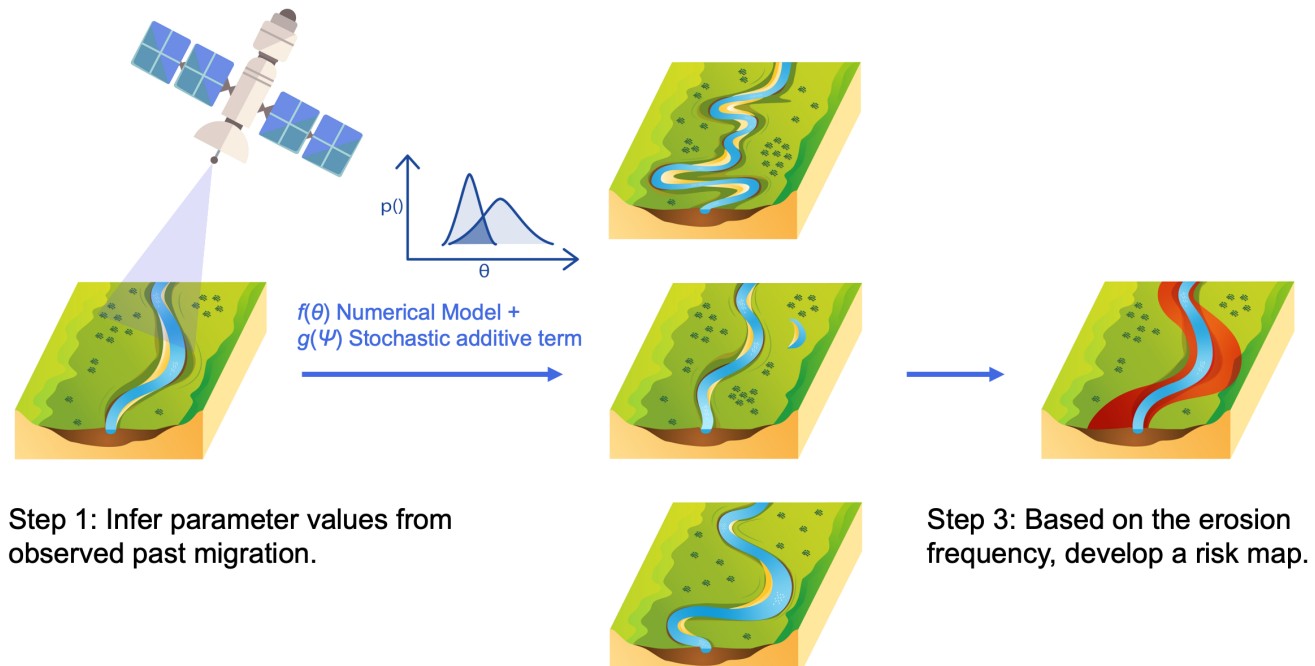

**Figure 3.** Illustration for the generation of a risk map using parameter distribution inferred from satellite observations (see algorithm 1).

---

**Algorithm 1** Generating risk map for river migration.

---

**Require:** Initial channel geometry $\mathcal{C}_0$

**Require:** A stochastic river-migration model $(f + g)$ that gives final channel geometry: $\mathcal{C}_t = \{(x, y) \in \mathbb{R}^2 \mid f(\theta, t) + g(\psi, s), \mathcal{C}_0\}$

**Require:** Posterior parameter distribution $p\left(\theta, \psi \mid \mathcal{C}_t^*, \mathcal{C}_0^*\right)$ [* denotes past observational data used for inference]

1: Generate samples from $p\left(\theta, \psi \mid \mathcal{C}_t^*, \mathcal{C}_0^*\right)$

2: Run the stochastic model $f(\theta, t) + g(\psi, s)$, using the generated samples, with $\mathcal{C}_0$ as the initial geometry (Figure 2)

3: Count all the times the migrated river $\mathcal{C}_t$ crosses a pixel within the meander belt (method: Figure 4)

4: Normalize the count with the number of Monte Carlo runs to get the erosion probability =0

---

## 2.4 Generation of risk maps of erosional hazard due to river migration

The goal of this study is to devise a framework that helps understand and quantify the risk of each pixel of land getting eroded within a timeframe by the evolving channel. Once we have the inferred model parameter distribution from the observed channel migration, we can use it to perform Monte Carlo simulation and generate multiple channel evolutions, the spread of which incorporates both parametric and model-structure uncertainty. To create a pixeled risk map over the alluvium, we need





to count the number of evolved channels that crossed a pixel. The ratio of the number of channels crossing a pixel $(x, y)$ to the total number of simulation runs $n$ gives us the probability of channel erosion, $p$, within the next t years at that location.

$$p = \frac{1}{n} \sum_{i=1}^{n} \mathbb{E}_i(x, y) \qquad \text{where} \qquad \mathbb{E}_i(x, y) = \begin{cases} 1 & \text{channel } i \text{ crossing the pixel at } (x, y) \\ 0 & \text{otherwise} \end{cases} \qquad (11)$$

However, it is not possible to uniquely identify the number of times an evolving 1D curve passes a point in a 2D plane from its initial and final geometry unless some more constraints are defined on the curve evolution. This is what we term as the counting problem of evolving channels. Nonetheless, we argue there are some unique properties to channel migration that can allow us to determine whether a channel has crossed a spatial point by simply comparing its initial shape and the final shape. This greatly reduces the computational burden of estimating erosion probabilities from Monte Carlo runs, as we are not

burdened with tracking the shape of the channel at every intermediate evolutionary timestep. The three constraints, enforced by the physical and empirical behavior of rivers that makes this counting possible are: (1) Channels flow downstream, therefore, at scales of multiple bends, they tend to enter the rectangular alluvial frame for which we want the risk map on one edge and leave on a different edge. (2) Channels evolve outwards on their convex side. This excludes curve evolution where a curve moves in one direction and then backtracks some of its paths later. These two constraints on the channel evolution help us come up with

a counting algorithm based on the initial and final position of the channel:

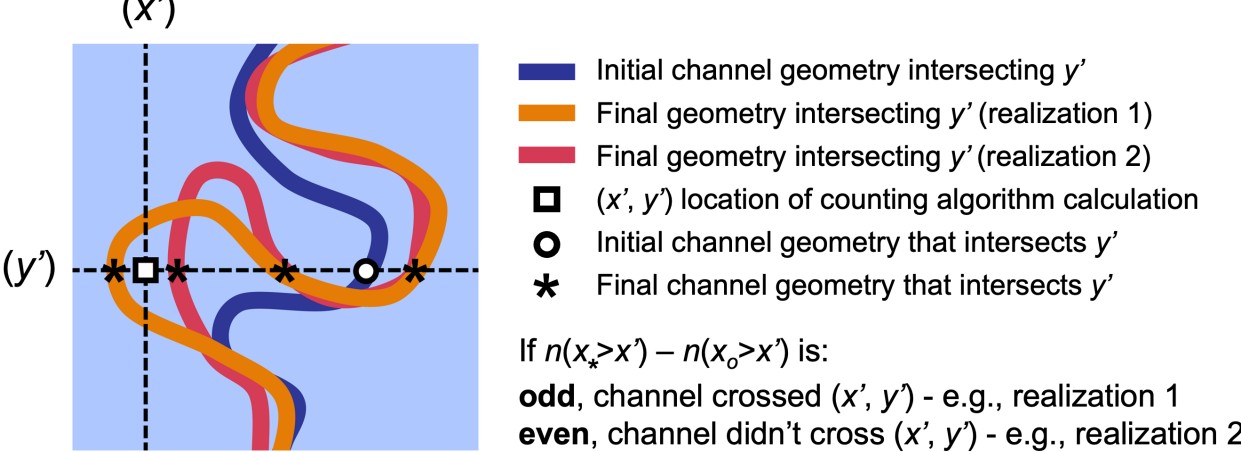

**Figure 4.** This figure illustrates the algorithmic counting scheme that allows us to establish whether a channel eroded past a geographic location (x',y') by analyzing its initial and final geometry. Here n(x>x') is a function that gives back the number of intersecting channel points on y' with an x value greater than x'.

Once these constraints are assumed, Figure 4 provides the method to establish whether channel geometry has evolved over a point. Corresponding to each pixel $(x', y')$ in the alluvial frame of interest, we draw a horizontal line and count how many





times the initial and final geometry of the channel crossed the line $y = y'$. We then divide the number of crossings between the left side and the right side of $x'$. If the number of crossings on either side has increased by an odd number, then the river has crossed the point $(x', y')$; Otherwise, it hasn't (Fig. 4). We repeat this process for all the pixels in the frame. And do this for all the Monte Carlo simulation runs. We finally do the counting of the channel crossings, and after normalization with the total runs, we get our geomorphic risk map, encoding the probability of erosion at each spatial location for a given time horizon (Fig. 3; algorithm 1).

In this work, we have not taken into account the width of the channel and only take into account the centerline of the channel. This is because incorporation of the width poses additional computational challenges. For example, the optimal methods to sequentially assign each on the center line to specific points on the bank have not been identified yet. Also, as our risk estimation algorithm is based on comparing the static configurations of initial and final channel geometries, it is not straightforward to identify which one of the two banks is creating the risk. However, the resolution of these challenges is not key to the assessment of our risk map generation framework, and therefore we leave it as a question to be researched further.

## 2.5 Performance assessment of generated risk maps

Unlike evaluating the performance of a deterministic forecast, where observed system realization is compared to the prediction using a distance metric (e.g., root mean squared error or correlation coefficient), evaluating probabilistic forecasts is more challenging as the prediction is available as probabilities. Here we describe a metric that can allow us to compute the performance of the probabilistic risk map. We also define the same metric over a subset of pixels to show the performance in avoiding false negatives. We start by defining the root mean squared departure (RMSD), where the departure is the difference in the probability value assigned to a pixel and the actual observed erosion in that pixel.

$$RMSD = \sqrt{\left( \sum_{i=1}^{n} (O_i - P_i)^2 \right)} \tag{12}$$

where we call the error metric as the root mean squared departure ($RMSD$). 'Departure' is the difference in the probability value assigned to a pixel and the observed erosion in that pixel. $O_i$ represents the observed value or erosion (0 or 1) for pixel $i$. $P_i$ represents the probability assigned for pixel $i$ by the probabilistic model (0 to 1) or the deterministic model (0 or 1). And $n$ is the number of pixels in the alluvial frame. Using the definition of Eq. 12, we can define a more specific metric that gives us the forecast performance for pixels where actual erosion occurs. This metric emphasizes the ability of the forecast to avoid overconfidence that can lead to false negatives.

$$RMSD_{eroded} = \sqrt{\left( \sum_{i=1}^{n'} (O_i - P_i)^2 \right)}, \qquad \text{where} \qquad i \mid O_i = 1 \tag{13}$$



## 2.6 Case study

To test our model, we focus on the Ucayali River in the Western Amazon Basin. The Ucayali is a single-threaded, meandering river that runs from the Andes through the Peruvian Amazon and eventually joins with the Marañón River to become the
275 Amazon River. The gauging station at Requena, Peru reports a mean annual discharge of 12,100 m³s⁻¹ from 2000-2015 (Santini et al., 2019). Within the reach of interest, the floodplain is unconfined, and the channel migrates rapidly across its floodplain, averaging 36 m yr⁻¹ of lateral migration (Constantine et al., 2014; Schwenk et al., 2017; Schwenk and Foufoula-Georgiou, 2016). We selected the Ucayali River to demonstrate the utility of our framework model for both these reasons, allowing us to observe a large amount of channel migration over the period with available satellite imagery that is largely unaffected
by topographic constraints, which is something that the Howard-Knutson model is unable to account for (Fig. 5). We utilize extracted channel centerlines of the Ucayali River produced by Schwenk et al. (2017) from Landsat 5 and 7 imagery between the years 1985-2015. We crop the area of interest into multiple alluvial frames to focus on 1, 2, 3, and 4 bends along a reach of approximately 20 km. However, the aim of our analysis is to show the strengths and weaknesses of our proposed risk map-generating framework, and not on the case study per se.

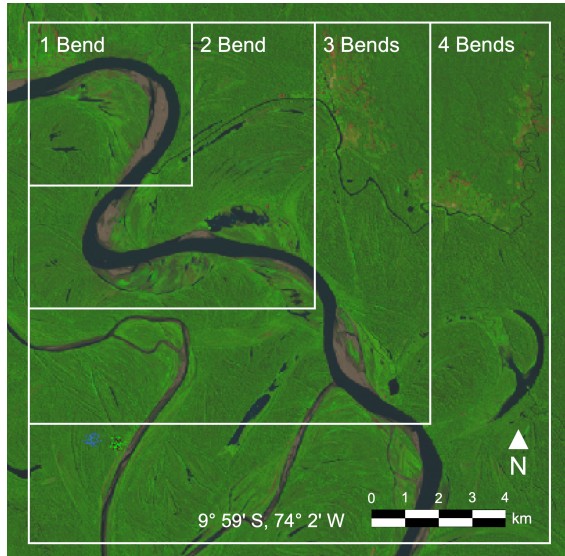

**Figure 5.** A Landsat image depicting the region of Ucayali River, with demarcated bends, used in this analysis to assess the proposed geomorphic risk estimation framework (image courtesy: the U.S. Geological Survey).

## 2.7 Model analysis using systematic simulation experiments

Our description of the stochastic geometric river migration model is put to the test in a systematic manner using a suite of simulation experiments. Through these model runs, we wish to understand the following aspects of the proposed mathematical description: (a) how well are we able to constrain the parameter values of the model using the satellite observations, (b) how





well are the risk maps produced using our stochastic description, in combination with inferred distribution of parameters, able
to provide improvements in reliability over deterministic descriptions, (c) and whether the underlying deterministic geometric
models, like Howard-Knutson, are able to capture the first-order dynamics of the migrating river system adequately.

The validity of the framework is tested using synthetic data in order to check the convergence of the posterior to the 'true'
parameter values that generate the observed river migration. Such controlled simulation experiments, where the parameter
values of the system are known a priori, help in ascertaining the ability of the inference algorithm and observational data
to constrain the system parameters. In the more detailed simulation experiments, we use the channel center lines from the
case study. We infer three parameter values of our stochastic description - migration rate $k_1$, friction factor $C_f$, and standard
deviation of the additive stochastic term $\sigma$ (Eqs. 4, 6, & 8). The MCMC simulations are run using 1, 2, 3, and 4 bends of the
Ucayali River as the initial state and evolve the state using our stochastic model with predetermined parameter values. The
multiplicative constants are used instead of the absolute value to give a comparable standard deviation to each parameter value.

The prior distribution of migration rate and the friction factor can be chosen based on expert opinion and specific character-
istics of the case study. In our case, where the river is rapidly migrating, we chose a relatively high value of the migration rate
constant. The specifications of the prior distribution that we used for our inference are as follows: uniformly distributed within
the range of [100, 500] for the migration rate constant, [0, 0.03] for the friction factor. For the standard deviation of the additive
error term, we use exponential distribution as its prior within the range [0.01, 10] and with the distribution parameter $\lambda$ as 1.
This gives preference to a smaller-magnitude additive error. We then use the initial and final channel geometry as synthetic
observations. The final geometry was generated using parameter values of 200, 0.01, and 0.1 (Eq. 2) for the migration rate
constant, the friction factor, and the standard deviation, respectively. For the inference exercise to be useful, it needs to learn
the value of these parameter values, going from a wide prior distribution to a narrow posterior distribution.

We then move on to study the effect of risk map generation on a real meandering river system using satellite observations.
In real systems, the underlying dynamics of river evolution is expected to deviate from the model that we deploy to forecast
it, thus putting the utility of our probabilistic framework to the test. We assume our stochastic description is the observation-
generating process, i.e., each simulation from the model, using a sampled parameter value from the posterior, is a forecast of
one possible course of river channel evolution, which is finally perturbed by a smooth random noise. A large ensemble of such
forward simulations, therefore, gives the aggregate statistics of our forecast. We conduct extensive simulations for inference
and prediction using the data from Ucayali. We infer parameters using the river profile in the year 1985 as the initial channel
geometry and the river profile in the year 1995 as the final channel geometry. We run numerical experiments in multiple batches
to understand the effect of (a) systematically increasing the number of bends on which the model parameters are inferred and
(b) and increasing the time horizon for which prediction is made. We systematically infer the model parameters using data
from 1, 2, 3, and 4 bends. We then predict the risk map for a time horizon of 5, 10, 15, 20, 25, and 30 years, thus capturing
the spatial dynamics of prediction uncertainty and confidence as the forecast lead time increases. In this analysis, we haven't
explicitly considered the with of the channel, and we simply evolve the centerline. As Ucayali is a fast-migrating river, there
is enough channel migration in multiples of 5 years that the proposed framework can be evaluated without incorporating with





width. However, for slowly migrating rivers or with small forecast time horizons, the orthogonal correction of channel width over the centerlines should be performed.

## 3  Results

In this section, we present the results from running our inverse and forward simulations using our stochastic description (Eq. 2). The results shed light on the ability of the likelihood function (Eq. 10) to help infer the unknown model parameter values using data from the past evolution of the channel. Finally, the forward simulations are evaluated based on various performance metrics for their corresponding risk maps (Fig. 3).

### 3.1  Ability to learn true parameter values from data

The results from the numerical simulations on synthetic data ascertain that the likelihood function on the summary statistic is able to retrieve the underlying parameter values that generated the data (Eq. 9 and Eq. 10). Figure 6 shows the convergence of the posterior to the underlying parameter values, with the highest probability mass gravitating towards 200 m/yr for the migration rate constant and 0.01 for the friction factor. We notice that increasing the number of bends used to infer these values does not necessarily reduce the standard deviation of the posterior distributions, as one would expect. This we attribute to more degrees of freedom that are added when the modeled river reach is extended. However, we learn a much narrower spread of parameter values compared to the assigned prior spread of [0, 500] and [0, 0.03] for the migration rate constant and friction factor, respectively (Fig. 6). As the EMCEE sampler uses multiple particle chains, they explore various regions of the parameter space. The initial parameter values in these EMCEE chains don't represent the underlying posterior and can, therefore, be excluded from the final samples as a burn-in phase.







**Figure 6.** The figure shows the posterior probability distribution of two parameters - migration rate constant and friction factor. The spread of the posterior captures the confidence in learned parameter values. Top row - (a) & (b) - shows inference using synthetic data; bottom row - (c) & (d) - shows inference using real data.

Results from the numerical simulation containing real observations (Fig. 6) show that the inference mechanism is able to learn a narrower posterior distribution from the observations. Similar to synthetic data, the reason behind longer river reaches not reducing the standard deviation of the posterior can be attributed to the increasing degrees of freedom of river migration





with scale. Additionally, this can be attributed to the fact that not each bend has the same value of the inferred parameter, and

adding more bends to the inference exercise doesn't necessarily make us confident about the value of the underlying parameter value of the process. Nonetheless, we do see the inferred parameter distribution using observations from 1, 2, 3, and 4 bends gravitate towards the same values of parameters (Fig. 6).

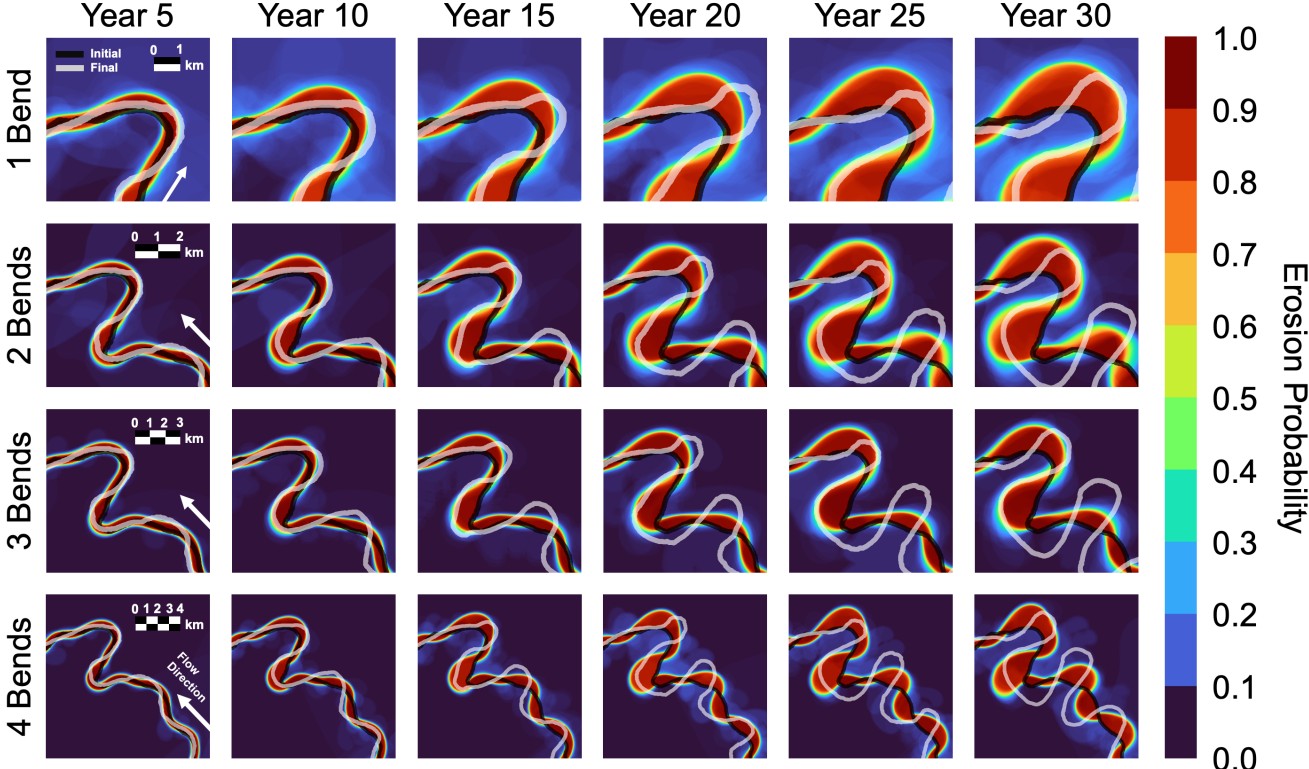

**Figure 7.** This panel of plots showcases risk maps generated for various forecast time horizons. Each row represents the simulation when the model is trained using 1, 2, 3, and 4 bends. The panels jointly represent the effect of increasing temporal and spatial scales of the forecast for river channel migration. Our framework is able to furnish spatial and temporal spread of erosion probabilities that deterministic paradigms are unable to provide.




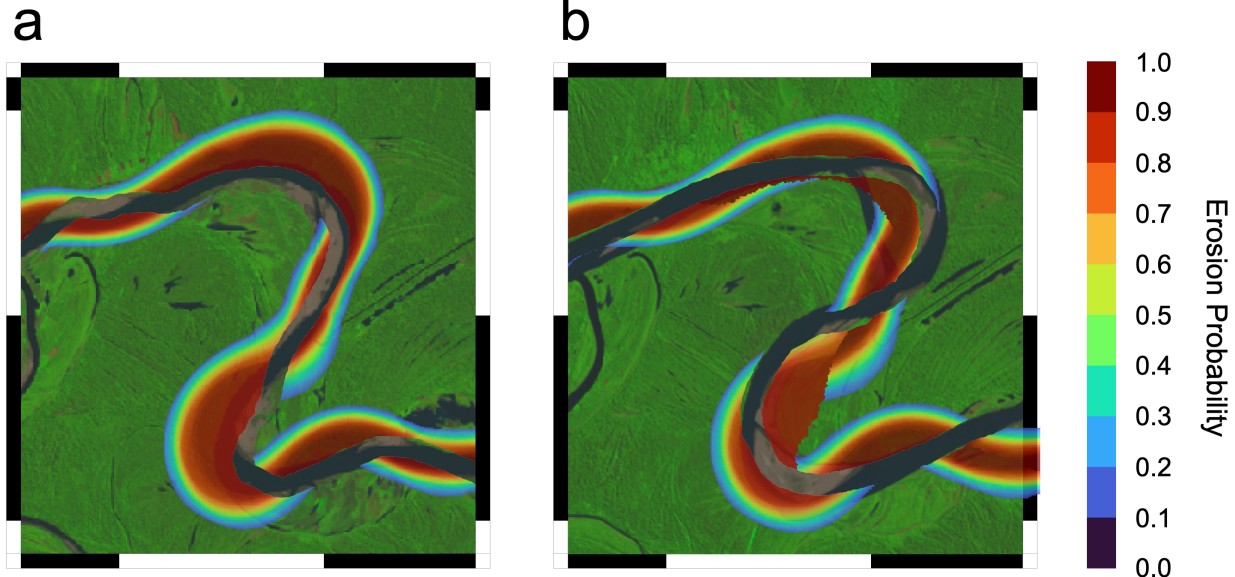

**Figure 8.** (a) Figure depicting risk map, over a Landsat image, generated using initial channel geometry in the year 1985. (b) The final migrated channel after a 10-year evolution. [Channel centerline - transparent].

### 3.2 Generating geomorphic risk maps from learned parameter values

After learning the posterior parameter distribution, we run the forward model (Eq. 2) using parameter samples from that distribution and employ our new framework to generate risk maps (Algorithm 1). The risk maps encapsulate the spatially distributed information about the probability that a geographical location in the alluvium will be eroded away by the migrating river within a certain timeframe (Fig. 7). As expected, we see that most of the risk is concentrated along sharply bending regions of the channel (Fig. 8). The risk map can be generated for as small a region as a stretch of a channel bend and all the way up to several meander bends. The extent of the risk map using a learned posterior distribution will be governed by the assumption that the underlying parameter values of channel evolution remain the same throughout the region. If the parameters of the stochastic model (Eq. 2) are believed to vary from one region to another, then one could either assign different model parameters to each region and infer them separately or define an explicit functional dependence of parameters with space. In that case, inference will be performed for the parameters of that functional dependence.

### 3.3 Effect of increasing forecast time horizon on risk estimates

We observe that the lengthening of the forecast time horizon increases uncertainty (Fig. 9). This uncertainty arises, among other factors, due to an accumulation of deviations in the real system from the modeled system. Besides, the nonlinear behavior of the river migration accentuates the effect of uncertainty in parameters and the initial channel condition for the final forecast. As we mentioned before, this uncertainty quantification should be encoded in the predictions, and we see that happening for





our probabilistic forecasts. The risk map captures this by assigning more pixels with a probability that is neither too high nor too low, e.g., between 0.25 and 0.75. therefore we see in Figure 7 that the probabilistic estimates are able to reproduce the decrease in certainty on whether the river will migrate across a location. The regions right next to the cut bank generally face a high probability of erosion. As the distance from the cut bank increases, the confidence in a region being affected by erosion within a time frame decreases. However, as the forecast time horizon increases, the probability of these places getting affected goes from low to moderate, thus increasing the uncertainty in the forecast. At the same time, forecasts become more confident about the erosion of pixels adjacent to the cut bank of the river. We use a probability value between 0.25 and 0.75 as a proxy of uncertainty and any probability value beyond this range as a proxy of certainty (Fig. 9). Notwithstanding the increase in uncertainty in particular regions, we are also able to become more certain that there will be erosion along the points that are adjacent to the migrating river. So both the rise in certainty and uncertainty of the forecast is captured by the probabilistic models, and the geographical spread of these probabilities allows us for risk-based decision-making. They also allow us to go out there in the field to make more detailed observations for locations of relatively high importance but where our forecasts are uncertain. Figure 8 shows how the actual channel profile may end up evolving over the forecast risk map. As we make a probabilistic prediction, the real observation is expected to sweep some areas that were labeled as high-risk but may miss out on some regions because of the system variability and model uncertainty. However, the overall performance of the probabilistic forecast comes out to be better than simply using one deterministic model run to evaluate erosional hazard (Fig. 10).

### 3.4 Effect of increasing spatial extent on risk estimates

The inference of parameter values should, in principle, be more precise as more observational data becomes available. However, in the case of river migration, there seem to be diminishing marginal gains in constraining parameter values as we include data from more bends. Figure 6 shows the posteriors for various simulation experiments. The posterior distribution does not become narrower when we use four bends to infer parameter values instead of one. This can be attributed to the fact that each bend might have a tendency to converge to a slightly different parameter value, and the overall posterior distribution ends up being wide when we force the model to have the same parameter values for each bend. However, when we only use a single bend for parameter inference, the learned distribution is narrow as the model can relatively easily fit to the shorter channel stretch. The result suggests that if the modelers have ample computational resources, they should use different migration rates and friction factors for each bend so that the most appropriate values can be learned in a spatially explicit manner. We also notice that trying to predict the river evolution for multiple bends using the same posterior distribution results in, to a certain degree, a relatively poorer prediction. While these results are qualified by their application to one case study, it appears that the prediction of risk due to erosion has better performance if it is done on a bend-by-bend basis.

Given the probabilistic nature of our framework, it enforces a forecast time scale on each case study for which the uncertainty in erosion will be maximum. Beyond that time scale, the river is expected to sweep its meander belt, and the forecast will start becoming more confident about the possibility of erosion, with the asymptotic behavior of all pixels in the meander belt getting an erosion probability of 1 for a large enough time scale. However, this will happen for long time scales of thousands of years. Conversely, for planning horizons of several decades, predicting the evolution of a river channel with high confidence is





difficult (Fig. 9). This is because we expect most of the pixels away from the cut bank will be assigned a probability between 0.25 and 0.75, which is a category signifying high uncertainty.

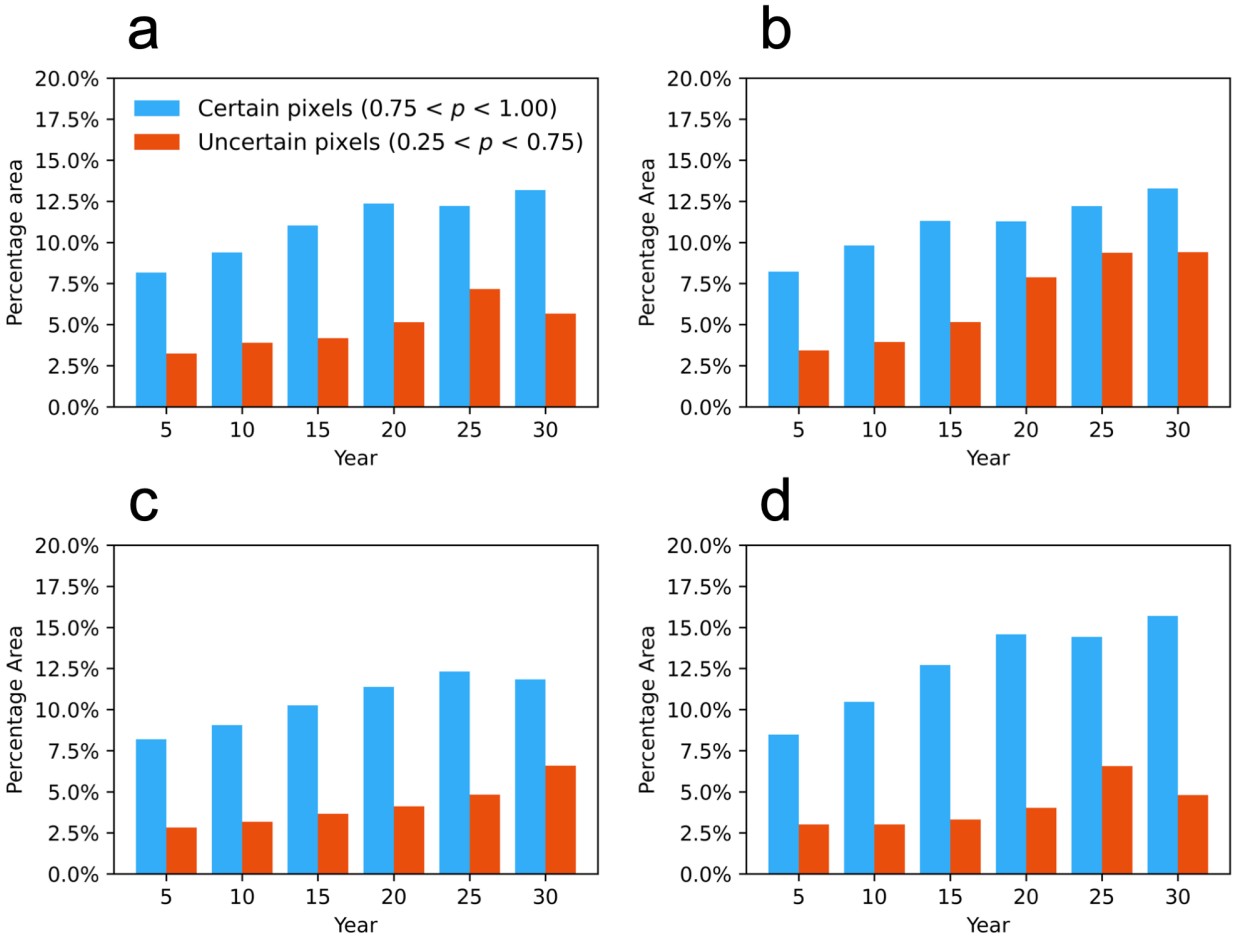

**Figure 9.** The effect of increasing forecast time horizon and spatial extent on the certainty and uncertainty forecasts (a - 1 bend; b - 2 bends; c - 3 bends; d - 4 bends). We classify certain pixels as the ones with an assigned probability of 0.75 or more. And uncertain pixels have an assigned probability in the range of 0.25 and 0.75. The figure shows that both the number of uncertain and certain pixels that face erosion grows as the prediction time horizon grows. This information is not furnished by deterministic forecasts.



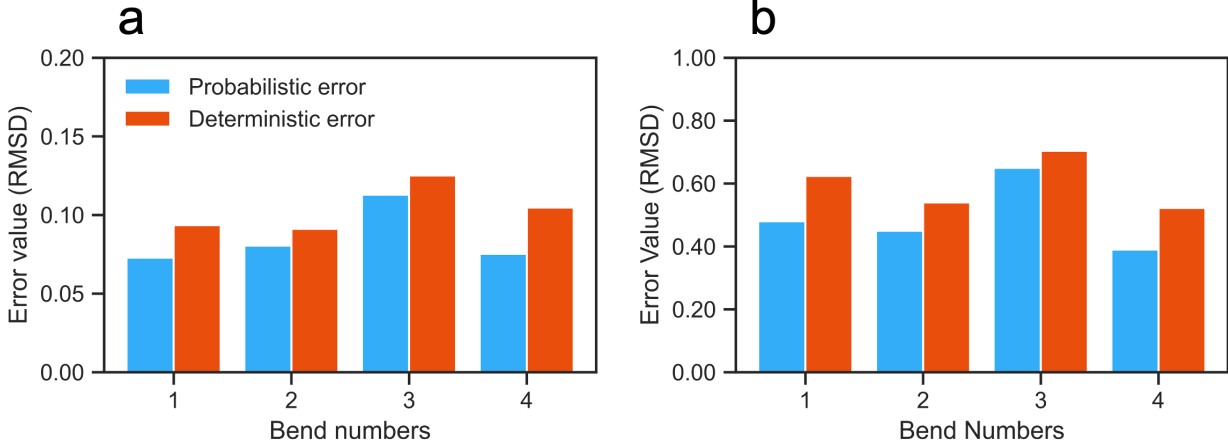

**Figure 10.** This shows the comparison of aggregate error (root mean square deviation) between probabilistic and deterministic forecasts. (a) RMSD, as defined in Eq. 12, provides the aggregate error in predicted risk and materialized hazard over all pixels.(b) RMSD, as defined in Eq. 12, provides the aggregate error in predicted risk and materialized hazard over the subset of pixels that end up actually being eroded. We can see the performance gains of probabilistic forecasts being represented as smaller error values.

## 4 Discussion

### 4.1 Reliability of probabilistic framework for risk estimation

The results show that uncertainty-aware forecasts are more reliable than deterministic forecasts, which do not explicitly report their uncertainties (Fig. 10). This reliability is attributable to two aspects of probabilistic models that their deterministic counterparts lack by construction. (a) The probabilistic forecasts avoid overconfidence in the forecast by assigning appropriate probabilities to regions that can potentially erode and avoid the strictly binary classification of deterministic paradigms. This is especially very valuable for risk analytics and prevents decision-makers from false negative classifications where risk is inadvertently underestimated (Fig. 10b). (b) As our probabilistic framework is based on monte carlo simulation using inferred parameter distribution, it allows the exploration of an entire region of the parameter space instead of a vector in that space. Therefore, the model is much more explorative in forward simulations and has a higher chance of capturing the eventual dynamics of the real channel with more fidelity.

### 4.2 Spatial distribution of certain and uncertain forecasts

Figure 7 also shows the effect of increasing the spatial and temporal extent of the forecast. Just as we see for short prediction time intervals, the forecast faithfully predicts the evolution of the river for short length scales, i.e., single bend (shown in row one of the panel in Fig. 7). We also notice that the forecast performance varies when we attempt to predict the evolution of different bends. This happens for both deterministic and probabilistic predictions; however, probabilistic models perform




better on the two error metrics (Fig. 10 a & b). This can partly be explained by the presence of a convolution integral in the Howard-Knutson model, which aggregates the curvature information from the upstream of a location. This aggregation can sometimes be informative and sometimes erroneously influence the local predicted rate of erosion at a bend. Such effects are mitigated when we learn the parameter values from single bends and only evolve the channel for the particular bend. Another factor that plays a role in forcing the same parameter values on various bends. As mentioned before, different bends will have different properties in relation to their bank substrate, channel slope, and aggregate hydrodynamics. Therefore, we speculate that learning parameters separately should improve this performance.

Nonetheless, one important feature of geomorphic uncertainty, which is expected qualitatively by forecasters but not possible to reproduce by deterministic paradigms, is that regions close to the river on the cut bank side are highly likely to be eroded, and regions farther away from the channel are unlikely to be eroded, and more intricate patterns of risk emerge in regions at a moderate distance away from the cut bank. This length scale is dictated by the prediction time horizon and shifts outwards as the prediction time horizon increases. Given our stochastic formulation, if two rivers have comparable parameter values and similar parametric and model-structure uncertainties, the bigger rivers with bigger bends will have a bigger zone of confidence for erosional hazard. Figure 7 shows that our probabilistic paradigm is able to capture this feature of uncertainty in the erosion forecasts of those spatial regions by assigning probabilities away from the extreme values of 0 and 1 (i.e., for example, between 0.25 and 0.75).

### 4.3 The value of Bayesian inference over other ad-hoc parameter inversion methods

Parameter inversion can also be performed in the deterministic paradigm, where a cost function, which generally encodes the aggregate error in the model prediction, is minimized by systematically exploring the parameter space. However, in highly stochastic physical processes, like channel evolution, the observations deviate from the first-order dynamics due to the aggregate effect of the neglected subprocesses. These deviations have a random character to them, hence confounding the inversion of parameters with uncertainty. Bayesian inference then becomes an attractive choice as a paradigm to learn distributions about model parameter values from observations. When the observations are very informative, we learn very narrow posterior distributions, and when the observations are not very informative, they don't constrain the posterior distributions to unjustifiably narrow regions of parameter space (Fig. 6). This formal updating mechanism allows for the incorporation of parametric uncertainty, which is otherwise unavailable when using deterministic parameter inversion techniques. Besides, regularization in the Bayesian framework is achieved by using informative prior distributions of parameter values. The ranges of these parameter priors can contain information about the physics of the system as well as expert knowledge about the geomorphology of the specific case study.

### 4.4 Geomorphic origins of stochasticity

The motivation to switch to stochastic models in a purely predictive framework is obvious – driven by the need to quantify uncertainty and have more reliable forecasts for river migration. However, from a scientific and mechanistic perspective within the context of fluvial landscape evolution, such modeling choices also seem to be the most natural. Scheidegger (1991) argues





that the noise in the river migration is too hard to track deterministically and, therefore, can be incorporated by considering
probability calculus. Similarly, given the system stochasticity, probabilistic descriptions of the bed load sediment flux have
been proposed (Furbish et al., 2012; Roseberry et al., 2012). In the same spirit, probabilistic cellular automata have been
used to understand and explain the behavior of emergent geomorphic dynamics due to rivers eroding and depositing sediment
over long time scales (Roberts and Wani, 2023; Martin and Edmonds, 2022). However, in these and similar discussions,
specifically related to the lateral migration of rivers, the forwards are studied in detail, but the inverse problem for such
stochastic formulations has not been studied. Our framework fills this gap and is able to provide a recipe by which we can
employ satellite imagery to invert the parameter values that drive the stochastic evolution of river channels while accounting
for parametric uncertainty.

## 4.5 The counting algorithm

The counting algorithm (Fig. 4), which is able to track whether a channel crossed a location in the meander belt by just looking
at the initial and final channel geometry, comes with a suite of assumptions about the migration dynamics that make this
otherwise indeterminate problem involving dynamic planar curves tractable. Without additional constraints, it is not possible
to know where the planar curve has been by simply looking at its initial and final shape. It could choose infinite paths to arrive
at the final shape. The additional constraints that go into making this algorithm are: 1) The river erodes along the cut bank
side. 2) The rectangular frame of the river under consideration is large enough such that the channel enters from one edge and
465 departs from a different edge. 3) The channel doesn't undergo a cutoff between its initial and final geometry. These assumptions
allow us to track regions that the river sweeps over during its migration. To the best of our knowledge, we don't know of any
mathematical literature on evolving planar curves that come with these special constraints. We, therefore, think this novel
proposal to track river migration, with its computational economy, is a useful contribution to the fluvial geomorphological
modeling community. The problems of cutoffs can be tackled by including the channel geometry just before the cutoff takes
place. This way, erosion can be calculated using the proposed algorithm using two parts of the channel evolution - pre-cutoff
and post-cutoff.

## 4.6 Adequacy of Howard-Knutson model

From the simulation experiments (Fig. 7) we notice that Howard-Knutson model is not always able to adequately represent
the first-order dynamics of channel evolution, as it neglects many complex subprocesses that influence the meander evolution.
To do monte carlo simulations, we need models possessing computational economy, which can run multiple times over using
available computing resources. However, these models nevertheless should be complex enough to learn behaviors present in
a variety of fluvial geomorphic settings. These behaviors can be captured in the parameter posterior distributions using the
observed migration in the past. However, the Howard-Knutson model does not come with enough parameters that can be said
to pack multiple model structures into it. For example, the model will not be able to start a migration if the river geometry is
480 already linear i.e., with not curvatures. Another limitation of Howard-Knutson model is that it only allows evolution along the
cut bank. While this is generally the case, some river sections can sometimes migrate in less restrictive ways, and a flood can



erode the point bar of a river. As our framework to generate risk maps is model agnostic, we suggest the forecasters should first
check the adequacy of their geometric models for capturing the first-order dynamics of the channel evolution.

### 4.7 Outlook

In this current formulation, we are using an additive stochastic term $g(\psi)$ to perturb the model output from a deterministic
numerical river migration model. This is one of the simple ways of accounting for model structure uncertainty. This formulation
can, however, be extended to include heteroscedasticity in time – where the contribution of the model structure errors to the
uncertainty in the final prediction will become even more pronounced as we forecast farther into the future. Although we see
the heteroscedastic effect in our current formulation, it is in its entirety attributable to the monte carlo simulations, i.e., the
490 uncertainty in the parameter values. This point can be elucidated by a thought experiment - if we can identify the parameters
narrowly enough, where the posterior becomes a Dirac delta function (or simply a vector in the parameter space), we will lose
heteroscedasticity in our forecasts as the only remaining stochasticity in predictions will be due to $g(\psi)$.

To address this limitation, further research is needed on more sophisticated formulations of extending deterministic river
channel evolution models into their stochastic counterparts. For example, we can a) make $g(\psi)$ into a function of time, where
its standard deviation grows in time and the time dependence is inferred from observations. b) Another more representative and
natural way would be to introduce stochasticity within the numerical model itself, such that the evolution of the river migration
is affected by noise in the geomorphic system. We also note that the additive stochastic error term sometimes generates very
large deviations from the predicted trajectories of the Howard-Knutson model. As such, stochastic formulations will not be
invertible in a straightforward manner within the Bayesian framework; we will have to employ approximate Bayesian compu-
500 tations. Notwithstanding these limitations, the current formulation of a smooth random function allows us to show the proof
of concept for our framework. We are able to form a stochastic model of river migration and infer the parameter values from
past observations of the river evolution. However, as mentioned before, real departures of river migration from our determin-
istic model come with a more detailed statistical structure compared to the truncated Fourier series with random coefficients.
This can be partly mitigated by inferring from the observations the number of sines and cosines that add up to form the error.
Additionally, we can allow the amplitudes of the departure to vary along the channel length. Such a flexible additive error
formulation will be able to produce better representations of a meandering river channel.

Another limitation of this work is that we have not included time horizons where cutoffs take place, as our counting algorithm
(Fig. 4) breaks down after cutoffs. We posit that the uncertainty in the predictions will grow markedly once the cutoff time
scales of channel evolution are reached, as cutoffs are the strongest source of nonlinearity in the system and can make this
system chaotic. Nevertheless, through our analysis, we intend to convey the value of switching to stochastic models of channel
migration for risk estimation and provide a systematic framework for achieving that switch.



## 5 Conclusions

In this paper, we propose a framework to extend deterministic numerical models for river migration into probabilistic models. Our framework allows for the incorporation of uncertainty in model structure and parameter values while also accounting for the effect of stochastic variations in the geomorphic systems. The framework is agnostic to the underlying numerical model used to capture the first-order dynamics of the migrating river. However, in this case study, we have conducted our analyses using a geometric model, with the formulation of Howard-Knutson, where the migration rate is dictated by a weighted sum of local and upstream curvature. From our analysis, we conclude:

– Deterministic models are uncertain and are additionally unable to capture the stochastic variability in the geomorphic system.

– Some of this uncertainty can be faithfully incorporated into the modeling exercise by using an additional additive stochastic term and using the inferred distribution of parameters instead of single values.

– We propose a recipe that uses smooth random functions as an additive stochastic term and then employs MCMC simulation to infer a posterior parameter distribution, which is conditioned on the observed migration in the river system.

– The recipe is shown to work in principle by using it on synthetic data (by inferring the parameter values that generated the data).

– We show that the method is able to identify regions in the vicinity of the river that are likely and unlikely to erode in the next $t$ years. Besides, the forecast is supplied in the form of a risk map, encoding both our confidence and uncertainty, which deterministic models are unable to do.

– We also propose a counting algorithm that enables to ascertain whether a river has crossed a spatial location by just looking at its initial and final geometry. We mention the assumptions and constraints of this algorithm.

– We see the Howard-Knutson model may not always be able to capture the first-order dynamics of the migrating river. We, therefore, suggest choosing the underlying deterministic model based on the geomorphic complexity of the case study.

*Author contributions.* OW and MPL conceptualized the study. OW developed the methodology, wrote the initial code, and wrote the manuscript. BN, with guidance from OW, built on the initial code, conducted detailed simulation experiments, generated and analyzed the results, and plotted the figures. OW, KBJD, and MPL supervised BN. MPL supervised OW and KBJD. All authors contributed ideas and provided feedback during the research phase. KBJD and MPL also contributed insights on the geomorphic modeling of migrating rivers. All authors contributed to the review and editing of the manuscript during the writing phase.



*Code and data availability.* https://github.com/braydennoh/StochasticRiverMigration

*Competing interests.* At least one of the (co-)authors is a member of the editorial board of Earth Surface Dynamics.

*Acknowledgements.* OW thanks the Swiss National Science Foundation for supporting with its Early Postdoc Mobility Fellowship, grant number P2EZP2 195654. MPL acknowledges funding from the Resnick Sustainability Institute at Caltech, NSF awards 2127442 and 2031532.




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

(R1)