# Peer review of "Geomorphic risk maps for river migration using probabilistic modeling - a framework"

_EGUsphere, 2023_

## Author Comment (AC1)

**Response to Reviewer 1 (Prof. Keith Beven)**

The edited manuscript has changes highlighted in blue, and all the line numbers referred to in our response correspond to the edited version. Thank you.

Reviewer comment 1: This paper represents a valuable first attempt to provide a practical approach to the probabilistic modeling of meander plan migration, with an application to satellite data for the Ucayali River in the Amazon basin.

As such, it is a bit outside my normal expertise, but I do have more experience in trying to apply probabilistic methods to deterministic knowledge in hydrological and other environmental applications. As such, one of the things that is lacking here is any recognition of the past discussions of epistemic and aleatory uncertainties in environmental applications and their consequences for model testing and uncertainty quantification. The main lesson learned, in fact, is that there is no right answer – what comes out depends on the assumptions made.

Authors' response: We are pleased to read that Prof. Beven considers our manuscript valuable. Thank you very much for reviewing our manuscript and for your suggestions.
To incorporate this comment, we have now further elaborated on and cited previous work done in the field on statistical treatment of uncertainties in landscape evolution and hydrosciences. We added five more references on uncertainty quantification, with one referring to models in earth sciences in general. (L48, L49). We have also rephrased part of the reviewer's comment and incorporated them into the introduction. "Environmental systems, like migrating channels, can be modeled by various types of governing equations - varying in their detail and complexity. These model structure choices dictate the performance of the model as well as the statistical properties of the residual errors. We, therefore, discuss the pros and cons of our statistical assumptions as well." (L67)

Reviewer comment 2: Here, I suspect that a professional statistician would be reasonably happy with the assumptions made since the problem has been shoehorned into a formal statistical framework (with consequent discussion in the

paper about the possibility of variability of parameters in time and space). But are those assumptions correct? A comment on Figure 6 suggests that "the parameters gravitate towards the same values" (L347). Even in the hypothetical case, where the assumptions are mostly me by definition this is surely not the case – there is a move of the migration coefficient away from the true value. Is the result therefore being biased by the likelihood function? The differences for the actual application are even more marked (the later discussion is more realistic in this respect).

Authors' response: Thank you for pointing this out. We agree that the likelihood function comes with assumptions about the statistical properties of errors. We mention that explicitly in the paper ( section 4.7: L493 - L521). Now we have added a line in the introduction and conclusions as well (L67 and L545).
"We also suggest further research into the extension of our additive stochastic term, enabling it, for example, to have variable statistical features in space and time."

Figure 6: Given we only have observations limited in space and time, the posterior will not necessarily have a mode at the underlying parameter value. Even when the assumptions about the distribution are correct - this is because a single trajectory of observed migration is a random sample. But our posterior faithfully captures the parameter value in its spread. To demonstrate this point, we include below a posterior of the mean of a normal distribution of two samples sample from that distribution. Theoretically, it can be shown that the standard deviation of the posterior shrinks with sqr( n), where n is the number of samples. And, as the reviewer points out, we have discussed the differences in the posterior distribution using real observations in detail in the discussion section. We now additionally mention this in the caption of the figure.

[Figure]

Posterior distribution generated using two observations - x1=-0.6, x2=1.46

Reviewer comment 3: So one of the questions (again with much discussion elsewhere but not here) is how far the uncertainty component is actually compensating for the model deficiencies and at what point should the underlying model be considered invalid (see the discussion of model invalidation in Beven and Lane, HP 2022, and references therein). In fact, each model run has its own set of residuals that will not necessarily have common structure or parameters. To assume that they have certainly simplifies the analysis – but it is again an assumption (not "by definition" as stated on L206 – other choices would be possible). So, in conclusion, I suggest that some revision of the paper is needed to reflect some of the issues raised above, both in querying the choice of assumptions as the methods are presented, and in the discussion (especially in how epistemic uncertainties are being formulated as if they are purely aleatory).

Authors' response: We have changed the wording of L213 to explicitly mention that we assume Gaussianity. As mentioned before, we make it explicit that the error description can be made more representative (section 4.7 and L542). We additionally include and cite insights from Beven and Lane, 2022, to drive home the point of rigorously testing and (in)validating environmental models.

I would very much like to see extension to more meanders and longer time scales (surely the data are available) as I suspect that this might reveal more limitations of the assumptions – but I accept that might not be possible.   This is already a useful first attempt at uncertainty estimation of such a problem. [As an aside, perhaps for future studies you might consider a limits of acceptability approach to model evaluation?]

Authors' response: We hope to publish another manuscript with details on the application of this method on longer timescales and more rivers. As the reviewer mentioned, for this manuscript, we wish to abstain from including more simulation experiments. We believe limiting ourselves to introducing and explaining the framework will help maintain thematic focus.

Some other comments:

Equation 2.    Theta should be included in g(), even if you then later assume independence.

Authors' response: Resolved. We mention this in L100.

L135.  The thing about epistemic errors including model structural errors due to oversimplification is that they are not necessarily systematic – that is what makes strong statistical assumptions often difficult to justify.

Authors' response:  We have now made this point more explcit in L143. "Model structure deficits can lead to a combination of systematic (underestimation of the migration rates) as well as random deviations from the real system response."

L141.  The equifinality thesis has quite a long history in hydrology (see e.g., Beven, 2009, Environmental Modelling – An Uncertain Future?)

Authors' response: We have included this citation.

  L154.   There is a lot of experience with model evaluations and uncertainty estimation of flood risk maps (some mentioned in the Beven and Lane paper)

Authors' response: We have included more citations, including the one mentioned by the reviewer.

L157. Follow some parametric distribution in the limit. You can assume that of course, but this is a nonlinear model subject to epistemic uncertainties so will not necessarily foloow in tie or space.

Authors' response: As the reviewer had cautioned before in his comments, so we have made this point explicit with various additional lines.

L374/5. Going out in the field. An interesting comment since you are not using a process model and you can get satellite images (and therefore explicitly quantify actual patterns of residuals) relatively frequently – so what would you actually measure? Might you might not better suggest allowing data assimilation in updating the forecasts (or will that be the next paper using more up to date images?)?

Authors' response: Thank you for the suggestion. We have now removed the mention of field visits and revised it to include data assimilation.

This could obviously be done in space too, working from bend to bend (L388ff) rather than as a spatially distributed inverse problem with all the interactions between bend parameter sets – since the best prior estimate of the distribution of parameters for each bend should be that of the upstream bend (unless there is information otherwise).

Authors' response: Thank you for this suggestion. We think this spatially sequential updating of parameter values from bend to bend would be an interesting hypothesis to test in future work. For this work, we have assumed that the same parameter values generate the observed migration for the entire domain of the channel.

L438. But in your case the observations are informative in the hypothetical case because the assumptions are consistent. In the real case, they are not, but the observations might still be informative – for example, in showing that your model is wrong (as suggested by Figure 8).

Authors' response: We have now revised our language to mention that we mean "informative" in the context of parameter estimation. And we agree that observation in general always carry information to validate or invalidate a model (L447).

**Additional references added to the manuscript:**

Beven, K.: A manifesto for the equifinality thesis, Journal of Hydrology, 320, 18–36, https://doi.org/10.1016/j.jhydrol.2005.07.007, 2006.

Beven, K. and Lane, S.: On (in)validating environmental models. 1. Principles for formulating a Turing-like Test for determining when a model is fit-for purpose, Hydrol. Process., 36, 2022.

Borgomeo, E., Hall, J. W., Fung, F., Watts, G., Colquhoun, K., and Lambert, C.: Risk-based water resources planning: Incorporating probabilistic nonstationary climate uncertainties, Water Resources Research, 50, 6850–6873, https://doi.org/10.1002/2014wr015558, 2014.

Büchele, B., Kreibich, H., Kron, A., Thieken, A., Ihringer, J., Oberle, P., Merz, B., and Nestmann, F.: Flood-risk mapping: contributions towards an enhanced assessment of extreme events and associated risks, Natural Hazards and Earth System Sciences, 6, 485–503, https://doi.org/10.5194/nhess-6-485-2006, 2006.

Caers, J.: Modeling Uncertainty in the Earth Sciences, Wiley, https://doi.org/10.1002/9781119995920, 2011.

Donovan, M., Belmont, P., and Sylvester, Z.: Evaluating the Relationship Between Meander-Bend Curvature, Sediment Supply, and Migration Rates, Journal of Geophysical Research: Earth Surface, 126, https://doi.org/10.1029/2020jf006058, 2021.

Gaull, B. A., Michael-Leiba, M. O., and Rynn, J. M. W.: Probabilistic earthquake risk maps of Australia, Australian Journal of Earth Sciences, 169–187, https://doi.org/10.1080/08120099008727918, 1990.

Liu, Y., Freer, J., Beven, K., and Matgen, P.: Towards a limits of acceptability approach to the calibration of hydrological models: Extending observation error, Journal of Hydrology, 367, 93–103, https://doi.org/10.1016/j.jhydrol.2009.01.016, 2009.

Neal, J., Keef, C., Bates, P., Beven, K., and Leedal, D.: Probabilistic flood risk mapping including spatial dependence, Hydrological Processes, 27, 1349–1363, https://doi.org/10.1002/hyp.9572, 2012.

Reichert, P.: Towards a comprehensive uncertainty assessment in environmental research and decision support, Water Science and Technology, 81, 1588–1596, https://doi.org/10.2166/wst.20

Reichert, P., Langhans, S. D., Lienert, J., and Schuwirth, N.: The conceptual foundation of environmental decision support, Journal of Environmental Management, 154, 316–332, https://doi.org/10.1016/j.jenvman.2015.01.053, 2015.20.032, 2020.

Slingo, J. and Palmer, T.: Uncertainty in weather and climate prediction, Philosophical Transactions of the Royal Society A: Mathematical, Physical and Engineering Sciences, 369, 4751–4767, https://doi.org/10.1098/rsta.2011.0161, 2011.

Wiel, M. J. V. D. and Darby, S. E.: A new model to analyse the impact of woody riparian vegetation on the geotechnical stability of riverbanks, Earth Surface Processes and Landforms, 32, 2185–2198, https://doi.org/10.1002/esp.1522, 2007.

Zargar, A., Sadiq, R., Naser, B., and Khan, F. I.: A review of drought indices, Environmental Reviews, 19, 333–349, https://doi.org/10.1139/a11-013, 2011.

---

## Author Comment (AC2)

**Response to Reviewer 2**

The edited manuscript has changes highlighted in blue, and all the line numbers referred to in our response correspond to the edited version. Thank you.

This new and valuable contribution sets out a probabilistic approach to the simulation of river meander migration that allows the generation of so-called geomorphic risk maps. The approach is potentially extremely valuable in that the derived risk maps offer a much more nuanced insight into the likelihood of different parts of the channel's floodplain being occupied. The paper is very well written and clearly argued throughout, so what follows might, for the most part, be regarded as minor queries/points of clarification rather than major critiques.

Authors' response: Thank you very much for reviewing our manuscript and for the favorable recommendation. We are very pleased to hear you find our manuscript clearly argued and grateful for your very helpful feedback and suggestions.

Reviewer comment 1: At Line 25 it is argued that there is some evidence that larger rivers (when averaged globally) migrate faster than smaller ones. However, the data on this is equivocal and it might be helpful to indicate a broader range of supporting (or conflicting) literature than just the recent analysis by Langhorst and Pavelsky. Part of the issue here is the qualitative nature of the term larger, alongside how rates of migration are actually defined. For example, an empirical data compendium assembled by Marco Van de Wiel has shown that, when normalized by their channel width, the rates of lateral migration of the largest rivers are surprisingly low (and often lower than 'smaller' rivers).

Authors' response: We agree that the jury is still out there, and there are competing hypotheses on the dependence of river migration on river width. We have now updated and qualified our sentence.

Reviewer comment 2: At Line 54, it could be useful for the reader to include some citations to highlight examples of previous risk-mapping approaches of the type referred to here.

Authors' response: Thank you. We have now included five references.

Reviewer comment 3: One of my more substantial critiques of this work concerns the introduction of the Howard-Knutson framework, which is the basis for the analysis that follows. This is initially introduced at L95, and I felt that it would be helpful to introduce here some of the limitations (including those identified in prior empirical work) of that approach, in particular examples of where the simply assumed relationship between curvature and migration breaks down. In fairness the authors do address these limitations towards the end of the work, but by deferring that discussion, the reader is left with a slightly false impression of the potential capabilities of the modelling framework. Given that one of the key advantages of a probabilistic approach is that it could potentially highlight incidences of unusual river behavior (especially behavior that is low probability but high consequence), then the exclusion of instances of channel migration that do not conform to the Howard-Knutson model, but which are known to occur in nature, is regrettable. It is, of course, very difficult to include all such instances in a single model, especially when the main aim of the paper is to highlight a new methodological framework. But I do feel that addressing this unavoidable difficulty head-on and early would be helpful to readers.

Authors' response: Thank you for pointing this out. We now make this point explicit in the introduction (L121) rephrasing concerns of the reviewers as: "For example, there are many instances when this simple assumed relationship between channel curvature and migration rate does not hold or other controls dominate the dynamics."

Additionally, we conclude by saying L541: "We see the Howard-Knutson model may not always be able to capture the first-order dynamics of the migrating river. We, therefore, suggest choosing the underlying deterministic model based on the geomorphic complexity of the case study."

Despite the limitations of this particular model, we would like to mention again that our framework is model agnostic. The $f(\theta)$, which represents a channel migration model, can be suitably chosen without loss of generality of our probabilistic framework.

L490: "As our framework to generate risk maps is model agnostic, we suggest the forecasters should first check the adequacy of their geometric models for capturing the first-order dynamics of the channel evolution."

Reviewer comment 4: Is it really the case (L117-118) that the aim is to capture only the most likely evolution and not the whole suite of possibilities/probabilities? The former feels much more limiting than the latter.

Authors' response: Thank you. We meant that's the limitation of using fixed parameter values in a deterministic modeling framework. Whereas our proposed framework explores the entire suite of possibilities. We have now changed the wording to reflect this clearly.

**Additional references added to the manuscript:**

Beven, K.: A manifesto for the equifinality thesis, Journal of Hydrology, 320, 18–36, https://doi.org/10.1016/j.jhydrol.2005.07.007, 2006.

Beven, K. and Lane, S.: On (in)validating environmental models. 1. Principles for formulating a Turing-like Test for determining when a model is fit-for purpose, Hydrol. Process., 36, 2022.

Borgomeo, E., Hall, J. W., Fung, F., Watts, G., Colquhoun, K., and Lambert, C.: Risk-based water resources planning: Incorporating probabilistic nonstationary climate uncertainties, Water Resources Research, 50, 6850–6873, https://doi.org/10.1002/2014wr015558, 2014.

Büchele, B., Kreibich, H., Kron, A., Thieken, A., Ihringer, J., Oberle, P., Merz, B., and Nestmann, F.: Flood-risk mapping: contributions towards an enhanced assessment of extreme events and associated risks, Natural Hazards and Earth System Sciences, 6, 485–503, https://doi.org/10.5194/nhess-6-485-2006, 2006.

Caers, J.: Modeling Uncertainty in the Earth Sciences, Wiley, https://doi.org/10.1002/9781119995920, 2011.

Donovan, M., Belmont, P., and Sylvester, Z.: Evaluating the Relationship Between Meander-Bend Curvature, Sediment Supply, and Migration Rates, Journal of Geophysical Research: Earth Surface, 126, https://doi.org/10.1029/2020jf006058, 2021.

Gaull, B. A., Michael-Leiba, M. O., and Rynn, J. M. W.: Probabilistic earthquake risk maps of Australia, Australian Journal of Earth Sciences, 169–187, https://doi.org/10.1080/08120099008727918, 1990.

Liu, Y., Freer, J., Beven, K., and Matgen, P.: Towards a limits of acceptability approach to the calibration of hydrological models: Extending observation error, Journal of Hydrology, 367, 93–103, https://doi.org/10.1016/j.jhydrol.2009.01.016, 2009.

Neal, J., Keef, C., Bates, P., Beven, K., and Leedal, D.: Probabilistic flood risk mapping including spatial dependence, Hydrological Processes, 27, 1349–1363, https://doi.org/10.1002/hyp.9572, 2012.

Reichert, P.: Towards a comprehensive uncertainty assessment in environmental research and decision support, Water Science and Technology, 81, 1588–1596, https://doi.org/10.2166/wst.20

Reichert, P., Langhans, S. D., Lienert, J., and Schuwirth, N.: The conceptual foundation of environmental decision support, Journal of Environmental Management, 154, 316–332, https://doi.org/10.1016/j.jenvman.2015.01.053, 2015.20.032, 2020.

Slingo, J. and Palmer, T.: Uncertainty in weather and climate prediction, Philosophical Transactions of the Royal Society A: Mathematical, Physical and Engineering Sciences, 369, 4751–4767, https://doi.org/10.1098/rsta.2011.0161, 2011.

Wiel, M. J. V. D. and Darby, S. E.: A new model to analyse the impact of woody riparian vegetation on the geotechnical stability of riverbanks, Earth Surface Processes and Landforms, 32, 2185–2198, https://doi.org/10.1002/esp.1522, 2007.

Zargar, A., Sadiq, R., Naser, B., and Khan, F. I.: A review of drought indices, Environmental Reviews, 19, 333–349, https://doi.org/10.1139/a11-013, 2011.